# Arsenite Exposure to Human RPCs (HRTPT) Produces a Reversible Epithelial Mesenchymal Transition (EMT): In-Vitro and In-Silico Study

**DOI:** 10.3390/ijms24065092

**Published:** 2023-03-07

**Authors:** Sonalika Singhal, Scott H. Garrett, Seema Somji, Kalli Schaefer, Benu Bansal, Jappreet Singh Gill, Sandeep K. Singhal, Donald A. Sens

**Affiliations:** 1Department of Pathology, School of Medicine and Health Sciences, University of North Dakota, Grand Forks, ND 58203, USA; 2Department of Biomedical Engineering, School of Electrical Engineering and Computer Science, University of North Dakota, Grand Forks, ND 58203, USA

**Keywords:** inorganic arsenic, bioinformatics, kidney, nephrotoxicity, EMT

## Abstract

The human kidney is known to possess renal progenitor cells (RPCs) that can assist in the repair of acute tubular injury. The RPCs are sparsely located as single cells throughout the kidney. We recently generated an immortalized human renal progenitor cell line (HRTPT) that co-expresses PROM1/CD24 and expresses features expected on RPCs. This included the ability to form nephrospheres, differentiate on the surface of Matrigel, and undergo adipogenic, neurogenic, and osteogenic differentiation. These cells were used in the present study to determine how the cells would respond when exposed to nephrotoxin. Inorganic arsenite (iAs) was chosen as the nephrotoxin since the kidney is susceptible to this toxin and there is evidence of its involvement in renal disease. Gene expression profiles when the cells were exposed to iAs for 3, 8, and 10 passages (subcultured at 1:3 ratio) identified a shift from the control unexposed cells. The cells exposed to iAs for eight passages were then referred with growth media containing no iAs and within two passages the cells returned to an epithelial morphology with strong agreement in differential gene expression between control and cells recovered from iAs exposure. Results show within three serial passages of the cells exposed to iAs there was a shift in morphology from an epithelial to a mesenchymal phenotype. EMT was suggested based on an increase in known mesenchymal markers. We found RPCs can undergo EMT when exposed to a nephrotoxin and undergo MET when the agent is removed from the growth media.

## 1. Introduction

The tubular epithelium of the human kidney has the capacity to regenerate, repair, and re-epithelialize in response to injury by various insults. In the human kidney, a population of resident cells with progenitor characteristics, identified by the PROM1 stem cell marker, were localized to the Bowman’s capsule, proximal tubules, and the inner medullary papilla [1,2,3]. The number of cortical PROM1-expressing tubular cells increased in patients with acute renal injury [4]. Further studies have shown renal epithelial cells co-expressing PROM1 and CD24 have the capacity to participate in the regeneration of renal tubule cells [5,6,7,8,9]. Cultures of human renal epithelial cells that co-express PROM1 and CD24 also display features expected of RPCs; such as spheroid formation, ability to undergo adipogenic, neurogenic, osteogenic differentiation, and form tubule-like structures on Matrigel. These cells provided a potential model to define the mechanisms underlying the progenitor cell’s ability to participate in renal epithelial cell regeneration. However, these cultures were shown to possess two cell types, one co-expressing PROM1 and CD24 and another expressing only CD24 [10]. Subsequently, our laboratory identified an immortalized human renal proximal tubule epithelial cell line, RPTEC/TERT1, that also displays the two cell populations, one that cell sorting was used to isolate two new immortalized cell lines, one HRTPT that co-expresses PROM1 and CD24, and another, HRECT24T that expresses CD24 and no PROM1 [11]. The HRTPT cells expressed the features defined for RPCs while the HRECT24T cells displayed no features of RPCs [11,12,13]. The HRTPT cells provide a human cell culture model to determine if PROM1/CD24 co-expressing RPCs are susceptible to nephrotoxic agents. To the author’s knowledge, this is an unexplored area as regards RPCs.

Exposure of the HRTPR cells to inorganic arsenic (iAs) was chosen to test this hypothesis. Inorganic arsenic has an extensive distribution in the environment [14,15,16]. The kidney is the most susceptible of all organ systems to iAs exposure [17,18]. There is evidence that exposure to iAs is associated with renal disease. A study of 6093 participants from arseniasis-endemic areas in northeastern Taiwan showed a temporal relationship between arsenic concentrations ≥ 10 mg/L in drinking water and CKD (chronic kidney disease) [19]. The study also demonstrated a dose-dependent association between well-water arsenic concentration and kidney diseases. Other studies have also shown an association of iAs exposure with alterations in renal function and disease [20,21,22]. Thus, there is evidence from population studies that exposure to iAs is associated with renal disease, however, studies defining the concentration of iAs within the human kidney and specific cells of the nephron are rare. Accumulation is possible due to the presence of metallothionein (MT), a small molecular weight protein that is known to bind and sequester iAs within cells [23,24,25].

## 2. Results

### 2.1. EMT as a Function of Exposure of HRTPT Cells to iAs

Examination of the HRTPT cells exposed to iAs by light microscopy demonstrated a change in cell morphology at passage 3 when compared to cells unexposed to iAs (Figure 1A,B). When compared to the control, the iAs exposed cells were less closely packed, more disorganized, and had lost the ability to form “domes”. Domes are raised areas of the monolayer due to fluid accumulation and are a manifestation of vectorial active ion transport [10]. This change in morphology was evident for at least seven more serial passages (Figure 1C). The change in morphology suggested that the cells could have undergone an epithelial-to-mesenchymal transition (EMT). Further evidence for the possibility of EMT was provided in passage 8 by increased expression of ACTA2, TAGLN, VIM, and CDH2 and a modest decrease in expression of CDH1, (Figure 1D–H). The co-expression of PROM1 and CD24 mRNA was retained by the cells exposed to iAs but the expression was clearly reduced (Figure 1I–K). The change in morphology and gene expression by the iAs−exposed cells suggested global gene expression technology might assist in providing additional information if iAs were inducing EMT or a related mesenchymal alteration in the HRTPT cells. Lower concentrations of iAs (1.0 and 2.0 μM) elicited a similar shift in morphology and increased expression but at an extended number of serial passages (Appendix A).

The HRTPT cells exposed to iAs were assessed for their morphology and expression of the above genes when iAs was removed from the growth media. Light microscopic examination showed that by the 2nd passage the iAs− cells displayed a morphology similar to the HRTPT controls (Figure 2A,B) and by the 11th passage they were indistinguishable from the control (Figure 2C). The iAs− cells regained dome formation at both P2 and P11 following iAs removal from the growth media. The expression of the ACTA2, TAGLN, VIM, N-cdh, and E-cdh genes were also assessed and all except VIM, which was absent from the iAs−cells, showed a trend to return to control values (Figure 2D–H). The change in morphology and gene expression after the removal of iAs suggested that the cells might have undergone a mesenchymal-to-epithelial transition (MET). These results presented the opportunity to examine the global gene expression profile of a toxin-exposed renal progenitor cell that shows evidence of undergoing EMT and, upon toxin removal, the ability to undergo MET and return to an epithelial morphology. The ability of the iAs− HRTPT cells to dome is strong evidence of epithelial differentiation.

### 2.2. Global Gene Expression and Impacted Pathway Analysis

The above morphology and gene expression changes suggested that exposure of HRTPT cells to iAs induced EMT, and when iAs was removed, MET back to the morphology and gene expression of the control HRTPT cells. Global gene expression was employed to further explore the ability of HRTPT cells to undergo EMT and MET as a function of exposure to iAs. In this section, triplicates of all the samples, i.e., control cells (P0), iAs exposed cells at P3, P8, and P10 and P8 cells and iAs recovered P2, P11 included to determine the global distribution and to identify all possible relationships among different conditions. The first two components of the PCA plot, PC1, and PC2, carry 66.1% and 13.4% of the variance of the data and the P0 is far removed from P10 and P8 as compared to P2 and P11 (Figure 3A). Correlation analysis supported this relationship and demonstrated that P2 samples were most closely related to the P0 (Figure 3C). Differential gene expression analysis using Post hoc test with ANOVA identified 2478 probes varieties across all possible conditions (Appendix A). The hierarchical clustering of the top 100 differentially expressed genes was determined from these 2478 probes (Figure 3B). IPA was performed using the 2478 probe varieties across all possible conditions which identified hepatocellular carcinoma as the top hepatotoxicity function and renal damage as the top nephrotoxicity function (Figure 3D). GSEA analysis on 2478 probes identified thirty-six (36) upregulated pathways, and 368 downregulated pathways with nominalized *p*-value < 0.05 (Appendix A).

### 2.3. Gene Expression of HRTPT Cells Exposed to iAs

Global gene expression was performed between P0 versus P3, P8, and P10 (each group separately) cells and identified 247, 363, and 304 differentially expressed genes, respectively (Appendix A). For the three sets of differentially expressed genes, 106/247 genes were down-regulated and 141/247 up-regulated; 118/363 were down-regulated and 245/363 up-regulated; and 111/304 genes were down-regulated and 193/304 were up-regulated in expression (Appendix A). An intersection analysis of the three gene sets for commonality identified 167 common genes with 91 genes being up- and 76 genes down-regulated (Appendix A, Figure 4A).

When analyzing all the exposed samples together, i.e., iAs+ with respect to the P0, the first two components of PCA show 57.8% and 22.2% of the variance, respectively (Figure 4B) among the phenotypes. The variation between P8 and P10 is very narrow as compared to the P3 samples. A total of 280 probes (234 gene symbols) were found differentially expressed between these conditions (Appendix A) and a subset of the top 25 genes was determined from these differentially expressed genes (Figure 4C). A volcano plot shows the most significant upregulated and downregulated differentially expressed genes (Figure 4D) between iAs+ and P0.

### 2.4. Pathway Analysis of HRTPT Cells Exposed to iAs

GSEA was performed on the 167 gene-set selected as an interaction of significant genes identified from P0 versus each of the iAs exposed cells passage (i.e., P3, P8, P10). We found 37 upregulated, and 129 downregulated pathways that had a nominalized *p*-value < 0.05 (Appendix A). Again, 167 common gene set was also analyzed using Reactome and we found the down-regulate pathways were associated with signaling pathways, especially those associated with FGF (Appendix A). Other associated pathways such as PI3K, the RAF/MAP kinase cascade, and ERBB were also identified using Reactome. The analysis of the 76 down-regulated gene set using Reactome also identified IGF signaling as a pathway. The prominent pathways associated with 91 up-regulated gene sets were interleukin signaling (IL4, IL10, IL13, IL18) and chemokine receptors (Appendix A). An analysis of the 167 gene set by the Panther Classification System also identified signaling pathways as a prominent component (Appendix A).

In addition, we performed pathway analysis on a total of 280 probes (234 gene-symbols, Appendix A) were found differentially expressed between P0 and iAs+ conditions (*p* < 0.05 and FC < 0.5 or >2) (Appendix A). Of the 234 genes, we found 151 were upregulated pathways and 34 downregulated pathways with nominalized *p*-value < 0.05 (Appendix A). One of the upregulated pathways was the Hallmark Epithelial Mesenchymal Transition, a gene set with genes defining the epithelial-mesenchymal transition [26]. IPA analysis on the above 234 genes identified other significant pathways associated with exposure to iAs (Appendix A). A total of 533 pathways were demonstrated in this list out of which around 200 were under the *p* < 0.05. EIF2, Ferroptosis, and mTOR signaling were at the top of the list.

### 2.5. Progenitor Cell Properties of HRTPT Cells after Recovery from Exposure to iAs

The HRTPT cells recovered from iAs exposure were shown to retain the ability to differentiate, form nephrospheres and express PROM1 and CD24 in over 94% of the cells (Figure 5A–F). One noteworthy alteration in tubular differentiation was that the recovered cells demonstrated no significant change in the expression of aquaporin from control cells, but did exhibit a large increase in the expression of calbindin (Figure 5G,H). The osteogenic gene RUNX2 and neurogenic gene ENO2 showed a significant increase (Figure 5I,J); while neurogenic genes MAPT and NES showed no significant change in expression (Figure 5K,L) and adipogenic gene, PPARG showed a decrease in expression when compared to the control HRTPT cells (Figure 5M). The confocal images show the expression of AP, AQP1, and THP as the tubulogenic marker (Figure 5N–P); FN1 and CD10 as osteogenic markers (Figure 5Q,R); NF, β-tub, and GFAP as neurogenic markers (Figure 5S–U); and PPARγ and ADIPOQ as adipogenic markers (Figure 5V,W) expression in recovered cells.

### 2.6. Gene Expression Analysis of HRTPT Cells after Recovery from iAs Exposure

The HRTPT cells were assessed for their gene expression at the P2 and P11 passage following the removal of iAs. A comparison between the control HRTPT (P0) cells and the P2 cells demonstrated that 166 genes were differentially expressed between the two groups, with 30 upregulated and 136 down-regulated genes (Appendix A). A similar comparison between P0 and P11 cells demonstrated that 71 genes were differentially expressed with 39 up-regulated and 32 down-regulated (Appendix A). The common genes between the two gene sets were determined and 36 genes were common (Appendix A, Figure 6A), with 22 up and 11 down-regulated genes (Figure 6B,C). Three genes were found to differ in directionality between the P2 and P11, IGFBP3, NMNAT2, and CYFIP2 (Figure 6D).

PCA found significant separation between the two recovered samples with PC1-74.1%, as compared to the control with PC2-18.9% (Figure 6E). Differential gene expression identified 77 probes significantly different between the recovered cells versus the control (Appendix A). The heat map shows the top 25 differentially expressed genes along with a volcano plot of the results (Figure 6F,G). The gene expression analysis of control versus iAs recovered samples (iAs−) identified 426 probes that were differently expressed with *p* < 0.05 (Appendix A).

### 2.7. Pathway Analysis of HRTPT Cells Following Recovery from iAs Exposure

36 common genes were examined using Reactome and Panther databases. The Reactome database identified elastic fibers and pathways involved in the cell cycle and p53 interactions (Appendix A). The Panther database identified mostly signaling and regulatory processes. GSEA on 49 genes identified as differentially expressed (Appendix A) between the control and iAs−. Only two down-regulated pathways were identified at nominalized *p*-value < 0.05 (Appendix A). IPA on differentially expressed genes between iAs− vs. control (Appendix A), confirm p53 signaling as a canonical pathway (Figure 7, Appendix A).

### 2.8. Comparison of iAs Exposed HRTPT Cells and HRTPT Cells Following Recovery from iAs Exposure

An intersection of differentially expressed genes between iAs exposed HRTPT cells and HRTPT cells following recovery from iAs exposure found 9-genes of interest (Figure 8A). These genes included CLDN16, CTSE, PTH1R, CYFIP2, SCD5, LIX1, MFAP5, KCP, and SH2D1B. PC1 and PC2 were showing 51.8% and 27% variance in the data, respectively (Figure 8B). A total of 305 differentially expressed probes (280 gene-symbols) were identified between P0, P3, P8, P10 and P0, P2, P11 with 41 down-regulated and 264 up-regulated genes (Figure 8D, Appendix A). GSEA on the 280 genes (Appendix A) found 42 downregulated pathways, and 157 upregulated pathways with nominal *p*-value < 0.05 (Appendix A). IPA identified FGFR as a significant upstream regulator (Figure 9, Appendix A).

## 3. Discussion

The HRTPT cell line provides an opportunity to determine how a human renal progenitor cell responds to a nephrotoxic agent and its subsequent removal. The hypothesis is that nephrotoxin might alter the regenerative capacity of the RPCs to repair tubular damage. The results demonstrated that the HRTPT cells exposed to 4.5 µM iAs displayed those characteristics of a cell undergoing EMT as noted by a change to a mesenchymal morphology and an increase in expression of mesenchymal markers such as ACTA2 and TAGLN. It was also shown that the alteration in morphology and increased expression of smooth muscle actin alpha 2 and transgelin also occurred at lower levels of iAs exposure (1.0 and 2.0 µM), albeit at much longer times of exposure, providing evidence that results found with 4.5 µM iAs would translate to lower levels of exposure. Global gene expression was used to further analyze the EMT response when the HRTPT cells were exposed to iAs. GSEA of the common genes expressed from the comparison of P3, P8, and P10 compared to control identified the Hallmark Epithelial Mesenchymal Transition from the MSigDB as an upregulated pathway [26]. The global differently expressed gene set was examined to determine if the iAs treated cells were transitioned to myoepithelial (keratin expressing) or myofibroblast-like (vim expressing) cells. The common gene set did not show the differential expression of any keratin genes or the vimentin gene. To further explore this finding, the P3, P8, and P10 were examined separately for keratin and vimentin expression. This confirmed that vimentin was not identified as differentially expressed for any of the three passages. In contrast, KRT18 was increased in expression when P3 and P8, but not P10, were compared to the control. This provides evidence that the transition favors the myoepithelial cell. KRT18 has been noted to increase during tubular injury and approximately 20-fold in the early stage following human renal transplantation. [27,28] To the authors’ knowledge this is the first observation that a human RPC can undergo EMT.

Pathway analysis for the 167 gene set identified signaling pathways associated with FGFR2 and chemokine receptors and chemokines as those having strong significance. An analysis of the down-regulated 76 gene set identified strongly with signaling related to the FGFR2 pathway, while the 91 up-regulated gene set was more strongly related to chemokine receptors, chemokines, and related pathways. An interesting feature of the down-regulated 76 gene set is that it was identical for all three time points of iAs exposure. The 76 gene set included FGF 9 and FGF 13 in addition to the FGFR2 receptor. The FGFR2 receptor has been shown to protect against tubular cell death and acute kidney injury involving ERK1/2 signaling in models of renal ischemia and reperfusion [29,30]. The expression of FGF9 has been shown to maintain the stemness of renal progenitor/stem cells during renal development [31]. FGF9 also has an essential role in the development of mesenchymal components in cells and tissues [9,32,33]. The FGF13 is elevated in ischemia/reperfusion in concert with the FGFR2 receptor [29]. The FGF18 gene, the only up-regulated FGF gene in the 167 gene set, has seen only limited study in the kidney but has been shown to have increased expression in cisplatin-induced murine AKI36. In breast cancer, FGF18 has been shown to be involved in both cell migration and EMT [34]. Despite these findings, the individual components of the FGF pathway have seen a limited study in renal disease as it relates to agent-induced changes in EMT and MET recovery from those changes. Arguing against any cause-and-effect relationship between the FGF pathway and iAs-induced EMT is the observation that iAs increased the activation of ERK1/2 in the HRTPT cells that had been exposed to iAs and undergone EMT. This type of response suggests that the many other ligands that can influence the ERK pathways might be active in the iAs-induced EMT. The important observation is that ERK was activated during the EMT process.

The increase in chemokine receptors and their ligands might also play an important role during iAs induced EMT. The increase in expression of IL4, 10, 13, and 14 and the pathway identification of chemokine receptors bind chemokines would appear to have consequences for renal diseases in the human setting due to their role in immune responses and inflammation. Most studies on EMT involve its involvement in cancer progression. However, a role in renal disease was established a decade ago, indicating that renal epithelial cells could switch to a mesenchymal phenotype. [35,36,37] The involvement of EMT in inflammation [38], fibrosis [39,40,41], and wound healing [42] suggests a link between chemokines and EMT. The pathway analysis was consistent for the role of FGF and chemokines in the EMT of the HRTPT cells. The only pathway present in Panther, but not Reactome or David, was the WNT7a pathway. Wnt7a was increased in expression and provides some evidence for upregulation of the non-canonical Wnt-signaling pathway. [43,44] Both the canonical and non-canonical Wnt pathways have been linked with diabetic nephropathy [45]. Overall, this aspect of the study provides the first demonstration that a renal progenitor cell can undergo EMT when exposed to an environmental toxin. The time course of exposure provides a 167 gene set associated with the iAs induced development of EMT and corresponding 91 and 76 gene sets representing genes up- and down-regulated within the 167 gene set. These three sets of genes will be valuable in determining the expression, druggable targets, and prediction value in a wide variety of human renal diseases and other disease datasets associated with iAs exposure.

The second aspect of this study was to determine, once iAs exposure was stopped if the iAs-treated HRTPT cells would retain their mesenchymal properties. The results showed that by the second passage following iAs removal, the cells had regained an epithelial morphology indistinguishable from the control HRTPT cells. This represents the initial observation that RPCs that have undergone EMT due to toxin exposure, can undergo MET back to an epithelial morphology after toxin removal. This ability is consistent with the observation that renal epithelial cells arise during embryogenesis by mesenchymal-to-epithelial transition (MET) [46,47]. It was confirmed that the cells undergoing MET retained the co-expression of PROM1 and CD24 and the ability to form nephron spheres and undergo osteogenic, neurogenic, lipogenic, and tubulogenic differentiation. A difference in tubulogenic differentiation was found for the recovered cells in that they expressed high levels of calbindin and low levels of aquaporin whereas the control unexposed cells had the opposite expression levels. To further explore this finding, global gene expression was performed at 2 and 11 passages following iAs removal. Following the removal of iAs, the cells at both P2 and P11 showed a marked divergence from the iAs-exposed HRTPT cells at P8 and a return to an expression profile more in line with the control HRTPT cells. This was especially noticeable in passage 11. The common genes between the control HRTPT cells compared to both the P2 and P11 cells were 36. Of these 36 genes, 3 had a reverse in expression between the control and recovered cells (CYFIP2, IGFBP3, and NMNAT2). To determine if iAs exposure might have a lasting, or potentially permanent, effect on gene expression, a common gene set was identified for iAs exposed cells at P3, P8, and P10 with those unexposed through P11. One could speculate that epigenetic modification due to iAs exposure might produce long-lasting alterations in the genome after iAs removal. The 33 gene set did identify interactions with p53 and the cell cycle. The possible interactions with p53 and the cell cycle would be consistent with the long-term carcinogenic effects of iAs.

The obvious limitation of the study is that it is performed using cells in culture. The results will require validation in the human kidney.

## 4. Materials and Methods

### 4.1. Study Design

A flowchart of the study design is shown in the visual abstract (Figure 10).

### 4.2. Cell Culture

The isolation and serum-free culture conditions for the HRTPT cells have been previously described [10,11]. Confluent cultures of HRTPT cells were exposed to 4.5 µM iAs for 24 hrs and then subcultured at a 1:3 ratio in the continued presence of iAs until confluent. Following confluence, the cells were serially subcultured again in the presence of iAs until confluent. This was repeated for 10 serial passages. Additional cultures of iAs exposed cells at passage 8 were sub-cultured into iAs free growth media and continued in iAs free media for 11 additional passages.

### 4.3. Microarray Gene Expression

The gene expression profile was determined using the Clariom D Human Microarray (platform ID: GPL23126) on triplicate samples of control HRTPT cells (P0) and HRTPT cells exposed to 4.5 µM iAs for 3, 8, and 10 serial passages (named as P3, P8, P10) and after recovery (named as P2, P11) (GSE215904). Each sample has gone through quality control processing before downstream analysis. The total of 138,745 probes were analyzed for each sample under different conditions. Confluent cultures were used for the isolation of RNA.

### 4.4. Individual Gene mRNA and Protein Expression

The mRNA and protein expression of individual genes was determined using RT qPCR, western blotting, and flow cytometry as described previously [10,11].

### 4.5. Statistical Analysis

Statistical significance of genes was calculated by running t-tests [48] between pairs of groups such as all subset of passages with respect to P0, iAs+ (combination of P3, P8, and P10 passages) with respect to P0, iAs− (combination of P2, P11 passages) with respect to P0 and iAs+ versus iAs−. When comparing more than two groups, one-way ANOVA (Analysis of Variance) was performed. Scattered volcano plots were used to show the statistically significant genes with *p*-value < 0.05 and fold-change (FC) greater than two in both directions (up or down-regulation). The foldchange for each gene was calculated based on antilog-expression value between two phenotypic conditions. Most of the genes provided in different tables were selected based on *p*-value ≤ 0.05 with or without fold change (FC ≤ 0.5 or FC ≥ 2). A principal component analysis (PCA) was performed to test the distribution of replicates of passage samples and Pearson correlation was used to find the relationship between the different passage conditions as well as genes [49,50]. The first two components of PCA i.e., PC1 and PC2 were used to explain the amount of variance in the data according to phenotypic condition(s). Venn diagrams were used to demonstrate the union and intersection of genes in different conditions. The entire analysis was performed using R/Bioconductor.

### 4.6. Pathway Analysis

Different significant gene lists were examined using commercially available pathway tools such as QIAGEN Ingenuity Pathway Analysis (IPA) [51], as well as freely available Gene Set Enrichment Analysis (GSEA) [52], Reactome [53], Panther [54], and DAVID [55,56] software databases.

### 4.7. Gene Set Enrichment Analysis

Gene set enrichment analysis was performed on different gene sets identified through t-tests or ANOVA using *p* < 0.05 +/- FC < 0.5 or >2. In some cases, some genes with fold change > 4-fold were also included, regardless of the *p*-value to sets the relevance at the functional level. Probes were ranked according to their *p*-value and/or log2 fold change and all the probes without gene symbols were excluded as they cannot map with the different pathway databases. Ranked lists were used in the Gene Set Enrichment Analysis pre-ranked software for a minimum gene size of 5 with a maximum gene set size of 500 [52]. The MSigDB pathway database was used to identify enriched gene sets including Hallmark, C2, and C3 [57].

## 5. Conclusions

This study shows that human renal progenitor cells, in vitro, undergo EMT when exposed to a nephrotoxin and undergo MET upon toxin removal. In addition, this study identified several significant genes and pathways of interest associated with inorganic arsenic exposure/removal and their linkage with renal disease. These genes provide robust sets of biological functions that can be further validated to predict their association with different diseases. In this study, a variety of machine learning and statistical analysis approaches have been taken to establish in-vitro to in-silico concordance, including an unsupervised analysis of genes across different phenotypic conditions, which can be used as an analytical guideline for other researchers.

## Figures and Tables

**Figure 1 ijms-24-05092-f001:**
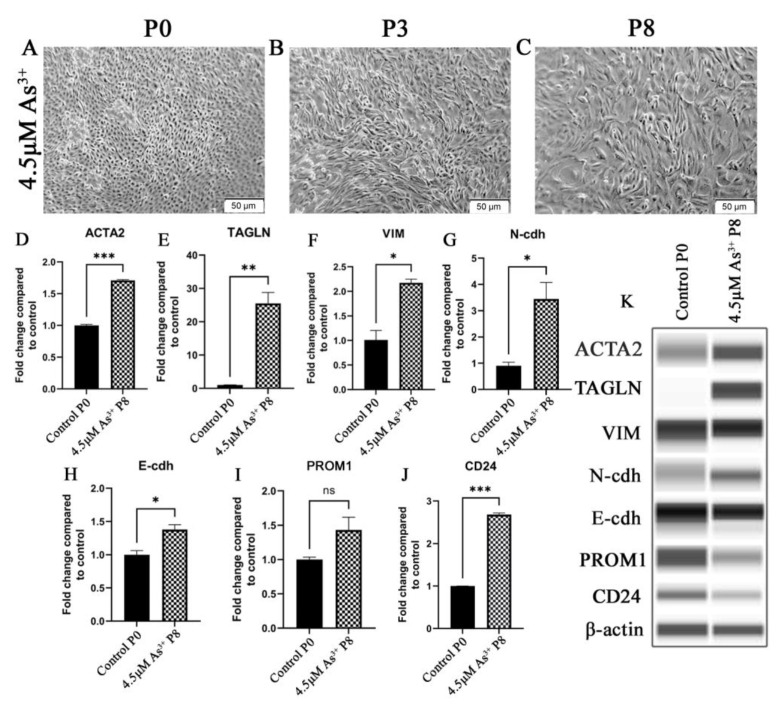
HRTPT cells exposed to iAs under light microscopy for (**A**) P0 (**B**) P3 (**C**) P8. Expression of P8 for epithelial-to-mesenchymal transition genes (**D**) ACTA2 (**E**) TAGLN (**F**) VIM (**G**) N-cdh (**H**) E-cdh and expression of (**I**) CD133 (**J**) CD24. (**K**) Western blot results confirmed protein level expression. Scale bar = 50 μm and Magnification ×10. * indicates *p* < 0.05, ** indicates *p* < 0.01, *** indicates *p* < 0.001.

**Figure 2 ijms-24-05092-f002:**
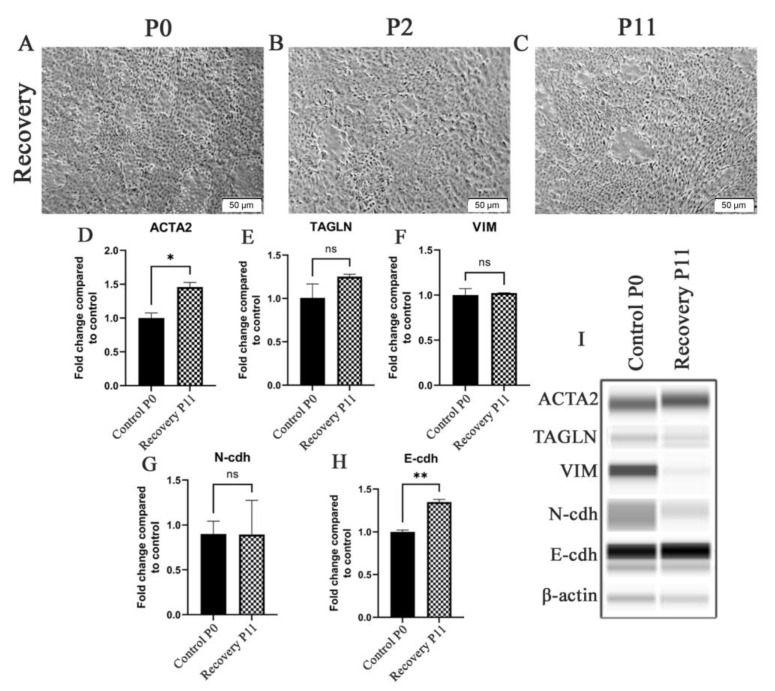
HRTPT cells exposed to iAs in recovery under light microscopy including (**A**) P0 (**B**) P2 (**C**) P11. Expression of (**D**) ACTA2 (**E**) TAGLN (**F**) VIM (**G**) N-cdh (**H**) E-cdh for recovery P2 and recovery P11. (**I**) Western blot showed protein levels in P2 and P11. Scale bar = 50 μm and Magnification ×10. * indicates *p* < 0.05, ** indicates *p* < 0.01.

**Figure 3 ijms-24-05092-f003:**
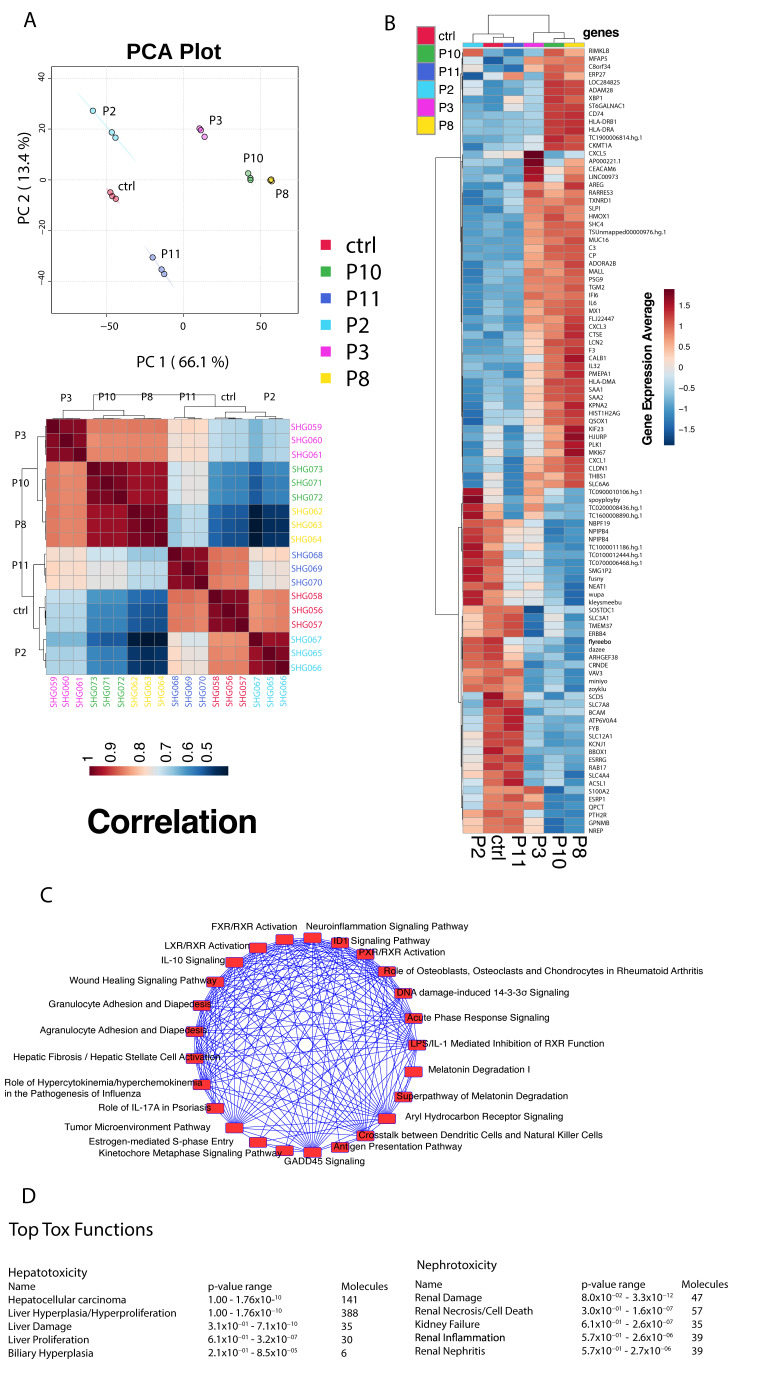
(**A**) Principal component analysis between the different passage conditions. (**B**) Hierarchical cluster analysis and heatmap of correlation. (**C**) Heatmap of gene expression averages for the top 100 differentially expressed genes identified through ANOVA analysis for the different passage conditions. (**D**) Top hepatotoxicity and nephrotoxicity functions from Ingenuity Pathway Analysis.

**Figure 4 ijms-24-05092-f004:**
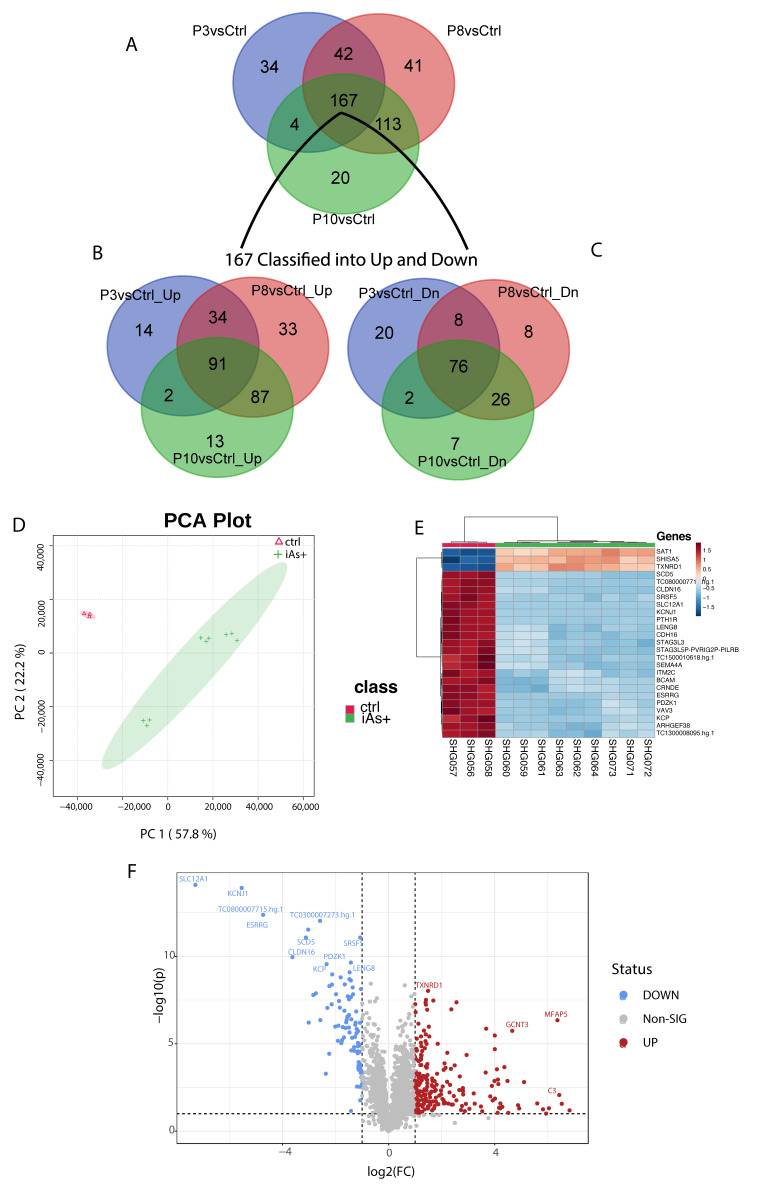
(**A**) Common genes among the differentially expressed gene set for P3 (246 gene set), P8 (363 gene set), and P10 (304 gene set) when compared to the control. (**B**) Common genes between the 167 gene set and the up-regulated genes from P3, P8, and P10 genes compared to the control. (**C**) Common genes between the 167 gene set and the down-regulated genes from P3, P8, and P10. (**D**) Principal component analysis of iAs+ cells with the control. (**E**) Hierarchical clustering of the top 25 differentially expressed genes for iAs+ cells with the control. (**F**) Significantly upregulated and downregulated differentially expressed genes based on the iAs+ cells with the control.

**Figure 5 ijms-24-05092-f005:**
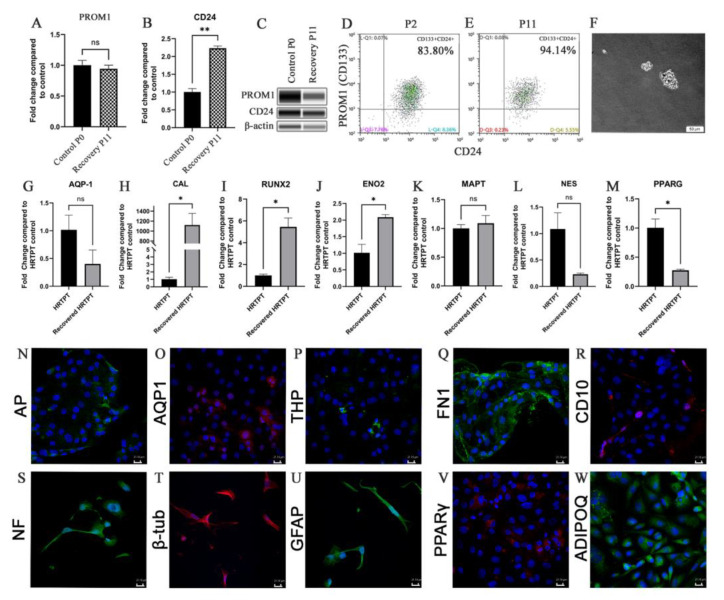
(**A**) PROM1 (**B**) CD24 expression for P0C and recovery passages P11AR. (**C**) Western blot of PROM1, CD24, and b-actin for P0 and recovery passages P11. (**D**,**E**) Flow cytometry expression of PROM1 and CD24. (**F**) Nephrospheres in HRTPT cells. Scale bar = 50 μm and Magnification ×10. (**G**) AQP-1 (**H**) CAL (**I**) RUNX2 (**J**) ENO2 (**K**) MAPT (**L**) NES (**M**) PPARG gene expression. (**N**) AP (**O**) AQP1 (**P**) THP (**Q**) FN1 (**R**) CD10 (**S**) NF (**T**) b-tub (**U**) GFAP (**V**) PPARg (**W**) ADIPOQ confocal microscopy, scale bar = 21.16 μm, magnification ×400. ns indicates no significance, * indicates *p* < 0.05, ** indicates *p* < 0.01.

**Figure 6 ijms-24-05092-f006:**
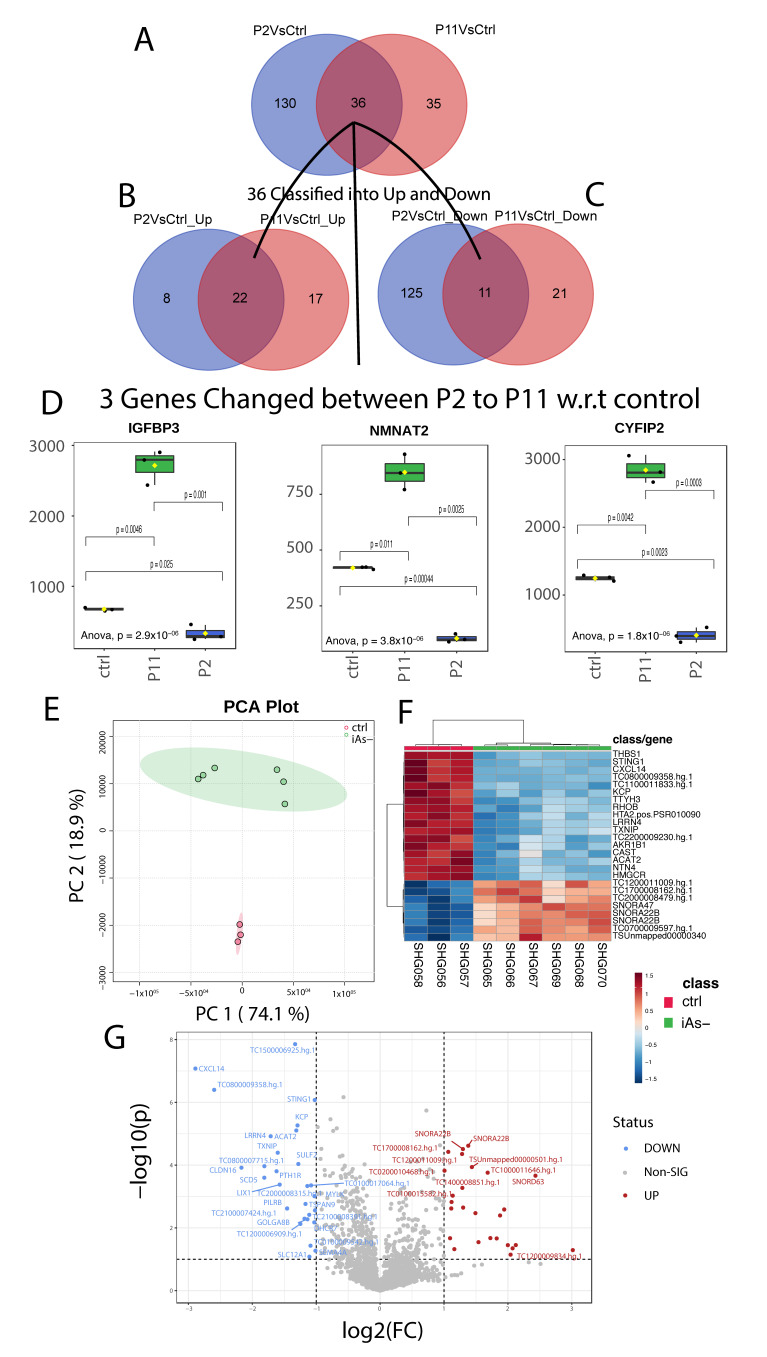
(**A**) Common genes among the differentially expressed gene set for P11 and P2 when compared to the control. (**B**) Common genes between the 36 gene set and the upregulated genes from P11 and P2. (**C**) Common genes between the 36 gene set and the downregulated genes from P11 and P2. (**D**) Log2 Foldchanges of genes with differing directions between P2 and the control, and P11 and the control. (**E**) Principal component analysis of iAs− cells with the control. (**F**) Hierarchical clustering of the top 25 differentially expressed genes for iAs exposed cells in recovery with the control. (**G**) Significantly upregulated and downregulated differentially expressed genes based on the iAs exposed cells in recovery with the control.

**Figure 7 ijms-24-05092-f007:**
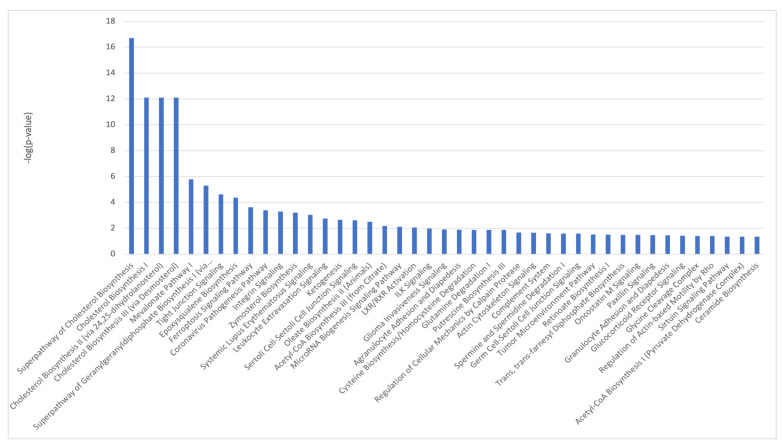
Ingenuity Pathway Analysis identified canonical pathways for control vs. iAs− conditions.

**Figure 8 ijms-24-05092-f008:**
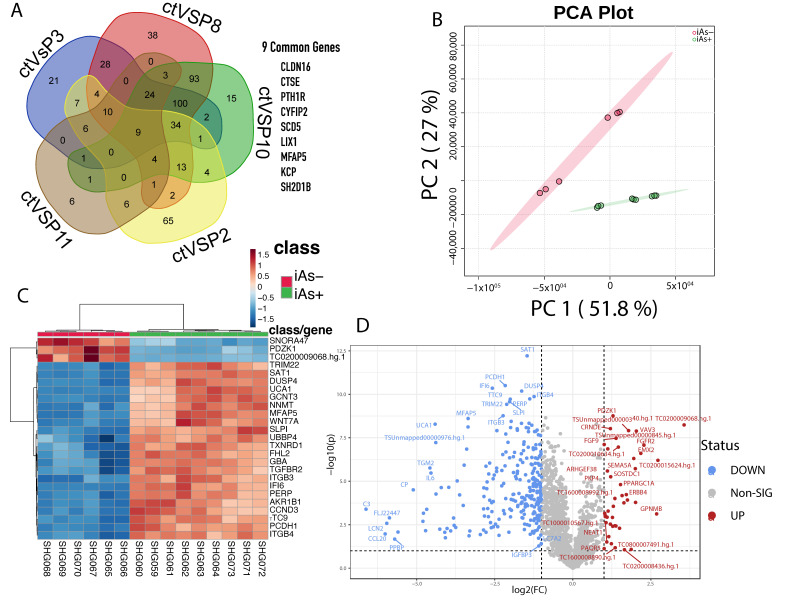
(**A**) Common genes between P3, P8, P10, P2, and P11 when compared to the control. (**B**) Principal component analysis of the two different conditions, iAs+ and iAs−. (**C**) Hierarchical clustering of the top 25 differentially expressed genes between the two conditions. (**D**) Significant upregulated and downregulated differentially expressed genes based on the iAs+ and iAs− conditions.

**Figure 9 ijms-24-05092-f009:**
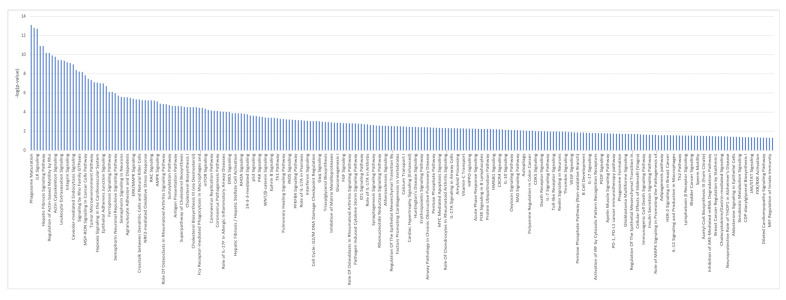
Ingenuity Pathway Analysis identified canonical pathways for iAs+ vs. iAs− conditions.

**Figure 10 ijms-24-05092-f010:**
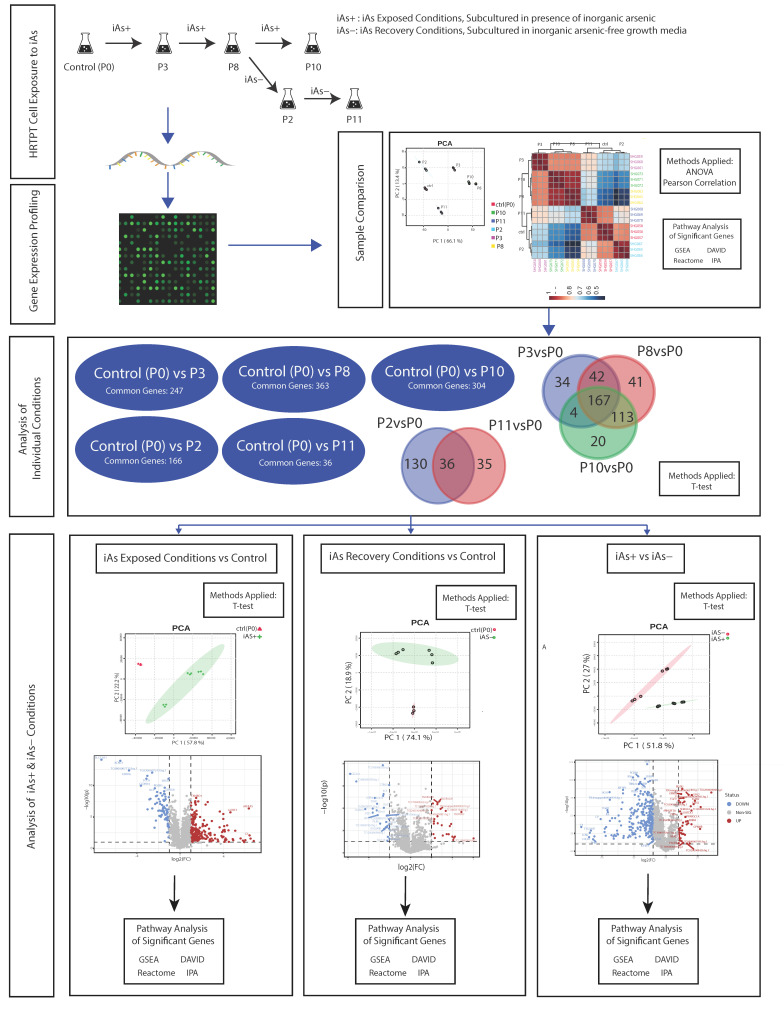
Flowchart of study design.

## Data Availability

The entire data have been submitted to GEO (accession numbers GSE215904). https://www.ncbi.nlm.nih.gov/geo/query/acc.cgi?acc=GSE215904 (accessed on 28 February 2023).

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
