# Peer review of "Arsenite Exposure to Human RPCs (HRTPT) Produces a Reversible Epithelial Mesenchymal Transition (EMT): In-Vitro and In-Silico Study"

_ijms, 2023, doi:10.3390/ijms24065092_

Round 1

Reviewer 1 Report

1. Overall, the paper was well-written and clear.

2. Abstract: Arsenite (iAs) à change to Inorganic arsenite (iAs)

3. Provide the first use of an abbreviation immediately before or after the expanded form. For example, at Line 59, what is “CKD”

4. Current western blot result in Figure 6 is not convincing. Suggest to upload the original western blot image. (In original images zip folder – no original western blot image)

5. Supp Figure 1 --> need to add scale bar in the microscopic images.

6. What is the conclusion of this study?

Author Response

Q1. Overall, the paper was well-written and clear.

Thank you for having taken your time to review our manuscript and provide us with your valuable feedback. Thank you so much for appreciation.

  1. Abstract: Arsenite (iAs) a change to Inorganic arsenite (iAs)

Agree. We have changed accordingly.

  1. Provide the first use of an abbreviation immediately before or after the expanded form. For example, at Line 59, what is “CKD”

Thank you for pointing this out. We agree with this and have incorporated your suggestion throughout the manuscript.

  1. Current western blot result in Figure 6 is not convincing. Suggest to upload the original western blot image. (In original images zip folder – no original western blot image)

I am really sorry about this. Initially, we provided the raw images to the editor via email and therefore, you were unable to see them. I have uploaded all raw images in a raw western blot images folder for your review. 

  1. Supp Figure 1 --> need to add scale bar in the microscopic images.

Thank you for pointing this out. We have added the scale bar and magnification to the legend of the images i.e. Scale bar = 50 μm. Magnification x10

  1. What is the conclusion of this study?

You have raised an important point here. We have added the following conclusion:

This study shows that human renal progenitor cells, in-vitro, undergo EMT when exposed to a nephrotoxin and undergo MET upon toxin removal. In addition, this study identified several significant genes and pathways of interest associated with inorganic arsenic exposure/removal and their linkage with renal disease. These genes provide robust sets of biological functions that can be further validated to predict their association in different diseases. In this study, a variety of machine learning and statistical analysis approaches have been taken to establish in-vitro to in-silico concordance, including an unsupervised analysis of genes across different phenotypic conditions, which can be used as an analytical guideline for other researchers.

Reviewer 2 Report

This present article by Singhal et al focused on how human renal progenitor cell lines (HRTPT) respond to Arsenite. Prolonged exposure of Arsenite results in epithelial to a mesenchymal transition. This could help learn more information on nephrotoxicity driven by Arsenite. Therefore, I principally support this work for publication. However, I have few minor suggestions to improve the work for publication.

Minor

  1. All light microscopy works should be displayed with scale bar and magnification (Fig 2,3,7F and s1). 

  2. Fig 7 N-W quantification is missing, also  scale bar and magnification need to be indicated. 

Author Response

All light microscopy works should be displayed with scale bar and magnification (Fig 2,3,7F and s1).

Thank you for pointing this out. We have added the scale bar and magnification to the legend of the images i.e. Scale bar = 50 μm. Magnification x10

Fig 7 N-W quantification is missing, also scale bar and magnification need to be indicated. 

Thank you for pointing this out. We have added the scale bar and magnification to the legend of the images. Scale bar = 21.16 μm. Magnification x400